# Recent Advances in Porphyrin-Based Systems for Electrochemical Oxygen Evolution Reaction [note 1]

**DOI:** 10.3390/ijms23116036

**Published:** 2022-05-27

**Authors:** Bin Yao, Youzhou He, Song Wang, Hongfei Sun, Xingyan Liu

**Affiliations:** Chongqing Key Laboratory of Catalysis and New Environmental Materials, College of Environment and Resources, Chongqing Technology and Business University, Chongqing 400067, China; yaobinpku@126.com (B.Y.); yzhectbu@163.com (Y.H.); wangsong@ctbu.edu.cn (S.W.)

**Keywords:** porphyrin, oxygen evolution reaction, electrochemical, heterogeneous catalysis, porous polymers, water oxidation, bifunctional oxygen electrocatalysis

## Abstract

Oxygen evolution reaction (OER) plays a pivotal role in the development of renewable energy methods, such as water-splitting devices and the use of Zn–air batteries. First-row transition metal complexes are promising catalyst candidates due to their excellent electrocatalytic performance, rich abundance, and cheap price. Metalloporphyrins are a class of representative high-efficiency complex catalysts owing to their structural and functional characteristics. However, OER based on porphyrin systems previously have been paid little attention in comparison to the well-described oxygen reduction reaction (ORR), hydrogen evolution reaction, and CO_2_ reduction reaction. Recently, porphyrin-based systems, including both small molecules and porous polymers for electrochemical OER, are emerging. Accordingly, this review summarizes the recent advances of porphyrin-based systems for electrochemical OER. Firstly, the electrochemical OER for water oxidation is discussed, which shows various methodologies to achieve catalysis from homogeneous to heterogeneous processes. Subsequently, the porphyrin-based catalytic systems for bifunctional oxygen electrocatalysis including both OER and ORR are demonstrated. Finally, the future development of porphyrin-based catalytic systems for electrochemical OER is briefly prospected.

## 1. Introduction

With the increasing population, the development of industry, the excessive consumption of fossil fuels, and the growing energy and environmental crises demand a sustainable energy supply. Water-splitting devices, metal–air batteries (i.e., Zn–air batteries), etc., are regarded as the new generation of energy conversion and storage techniques due to their advantages of low cost, environmental friendliness, and their use of a wide source of raw materials [1,2,3,4]. In the course of research, oxygen evolution reaction (OER), as one of the critical half-reactions, plays a vital role. However, owing to the theoretically high potential (1.23 V) and four-electron transfer processes, OER is essentially more difficult and complicated than hydrogen evolution reaction (HER), leading to sluggish kinetics [5,6]. Therefore, is the need to develop high-performance catalysts is inevitable. In particular, the Zn–air battery involves both OER and oxygen reduction reaction (ORR) half-reactions, so more stringent requirements for catalysts are needed [7].

Noble metal oxides (such as RuO_2_ and IrO_2_) have been demonstrated to have efficient electrocatalytic OER activities [8,9,10]. Nonetheless, their large-scale application has disadvantages of scarcity and a high cost limit. Inspired by biological systems that transition metal complexes actively participate in oxygen-involving reactions, first-row transition metal (such as Co, Ni, Fe, Cu, and Mn) complexes have especially attracted growing attention because of their rich abundances, cheap prices, and tunable multiple valences [11,12,13]. In numerous coordination complexes, metalloporphyrins play indispensable roles [14]. Porphyrins are a class of heterocyclic conjugated compounds containing four pyrrole subunits that are connected via methine groups. The planar conjugated structures render porphyrins to have favorable optical and electrical properties, such as the absorption of visible light and facilitated charge transfer routes. Porphyrin rings can coordinate with various metal ions and stabilize high-valent and low-valent species, which is of significant importance for the stability of key intermediates in catalytic cycles. The axial positions of porphyrins are mostly unoccupied, making the metal ions vulnerable to being attacked and ideal catalytic active centers. Meanwhile, the *meso*-positions are easy to be derived [15]. On one hand, electron-donating or electron-withdrawing substituents could be incorporated to regulate the charge density of central metal ions. On the other hand, the derivatized porphyrins can be used as building blocks to construct porous polymers, such as coordination polymers and porous organic polymers (POPs) [16,17,18,19]. The above-mentioned factors are meaningful to effectively modulate the catalytic features of porphyrins, making them excellent candidate catalysts for many processes, such as organic chemical reactions, ORR, HER, CO_2_ reduction reaction (CO_2_RR), and of course, OER [20,21,22,23,24,25,26,27].

Surprisingly, in comparison to ORR, HER, and CO_2_RR, few reviews are presented to systematically expound on the porphyrin-based systems for OER. Cao et al. published two review papers, but only porphyrin-based small molecules before 2016 [14] and a few porphyrin-based frameworks [28] are included. In recent years, a great number of porphyrin-based systems for OER have emerged, including both small molecule and porous polymer systems. In view of the great contributions of OER to sustainable energy, we realize a specific article that systematically summarizes the porphyrin-based systems for OER is very necessary. Moreover, there are several forms of devices that could conduct catalytic OER, including photochemical cells, photoelectrochemical cells, water-splitting electrolytic cells, and rechargeable Zn–air batteries (the latter two are shown in Figure 1) [28]. Nonetheless, for porphyrin-based photochemical cells and photoelectrochemical cells, photodegradation has always been an unsolved problem and extra components are also required, such as photosensitizers (i.e., [Ru(bpy)_3_]Cl_2_) as well as sacrificial electron acceptors (i.e., Na_2_S_2_O_8_) for photochemical cells [29,30,31,32,33] and noble metal complexes for photoelectrochemical cells [34,35,36]. The discussed existing problems limit the development of photochemical cells and photoelectrochemical cells. The devices based on electrochemical methods are considered to be more effective and environmentally friendly. Based on these considerations, this review summarizes the porphyrin-based small molecules since 2016 and porphyrin-based porous polymers, both amorphous and crystallized, for electrochemical OER.

## 2. Porphyrin-Based Systems as Electrochemical OER Catalysts for Water Oxidation

### 2.1. Porphyrin-Based Small Molecules for Electrochemical OER

Homogeneous water oxidation catalysts have received attention since they are important components of solar-fuel-generated devices. In 1994, Naruta reported the first series of manganese porphyrin dimers (Mn_2_DPs in Figure 2) for homogeneously electrocatalytic OER [37]. In the following two decades, homogeneous porphyrin-based catalysts have been greatly enriched. Many water-soluble non-noble metalloporphyrins, such as manganese porphyrin ([PMn(III)]Cl_5_) [38], cobalt porphyrin (TDMImP, TM4PyP, TTMAP) [39], and nickel porphyrin [40] (structures are shown in Figure 2), have been well-investigated as key components for electrocatalytic OER. However, perhaps not limited to metalloporphyrins, there are still critical problems to be resolved, such as high overpotentials, which means paying more energy costs.

The catalytic active sites of electrocatalysis are mostly located at the metal atoms of porphyrins and the modification of central atoms is the simplest and most effective method to regulate the overpotential. Copper plays critical roles in the formation and cleavage of the O–O bond in many biological processes. Inspired by this, in 2019, Cao et al. synthesized a water-soluble copper porphyrin (CuTMPyP in Figure 2) that showed low overpotential in neutral aqueous solutions for homogeneously electrocatalytic OER [41]. The onset overpotential was calculated to be 310 mV, which was lower than those of cobalt porphyrins and most reported copper complexes. Energy-dispersive X-ray spectroscopy (EDX) and SEM analysis confirmed the homogeneous and stable characteristics of electrocatalytic process since no heterogeneous nanoparticles were detected. Interestingly, except for the 4e water oxidation to produce O_2_, CuTMPyP also exhibited 2e water oxidation toward H_2_O_2_ in acid conditions. Further electrochemical experiments demonstrated that the formation of H_2_O_2_ was related to the second oxidation process of CuTMPyP. This, to some extent, reflected the reason for the low overpotential for electrocatalytic OER. In other words, the 1e-oxidized product of CuTMPyP could drive the formation of O–O bond, instead of involving the highly oxidized transition metal complex, leading to the decrease of overpotential.

The disadvantages for water-soluble small molecules serving as homogeneous electrocatalysts are obvious, such as weak interactions between catalysts and electrodes, low molecular utilization efficiency, and difficulty in recycling of catalysts. Several strategies are applied to develop heterogeneous molecular catalysts, and drop-coating of porphyrin-based small molecules onto conductive supports is the most straightforward means to realize heterogeneous catalysis. In earlier reports, several porphyrin-based small molecules (i.e., manganese porphyrins, cobalt porphyrins, and nickel porphyrins) and conductive supports (i.e., Li_2_O_2_ film, PEDOT, and carbon nanomaterial) were investigated for heterogeneously electrochemical OER [42,43,44,45,46]. Among these metalloporphyrins, cobalt porphyrins are proved to have better overall performances. Meanwhile, studies have also shown that, even for the same cobalt porphyrin, the heterogeneously catalytic performances vary greatly upon different conditions, such as the distinctive pH values of electrolyte solutions and the divergence of conductive supports. In addition to kinetic factors, the big differences in performances are also closely related to reaction intermediates and routes. Consequently, gaining deeper insight into the detailed catalytic mechanisms is essential for designing more efficient catalysts.

In 2017, Sun et al. investigated the intermediates of cobalt porphyrins loaded on a FTO electrode for electrochemical OER under borate buffer solution (pH 9.2) [47]. In their study, three distinctive cobalt porphyrin derivatives (CoTPP, CoN_3_O, and CoOEP in Figure 3) were designed and a critical synchrotron-based photoelectron spectroscopy (SOXPES) technology was utilized. Experimental results manifested that the real active species for three porphyrins were all a thin monolayer film of CoO_x_ (~5 × 10^−11^ mol/cm^2^) formed by the decomposition of cobalt porphyrins during electrocatalytic processes, which was highly transparent and showed high Faradaic yield (100%), as well as high turnover frequency (TOF 10 s^−1^). In 2019, Wan et al. combined UV−vis absorption spectroscopy and in situ electrochemical scanning tunneling microscopy (ECSTM) techniques to determine the intermediate of CoTPP for electrocatalytic OER on the Au(111) electrode surface under different pH values [48]. It was observed that the electrocatalytic system displayed positive performance in alkaline solution (0.1 M KOH) and the enhanced property was ascribed to a CoTPP−OH^−^ intermediate instead of CoO_x_ during electrocatalysis. Moreover, the CoTPP−OH^−^ intermediate was found to exhibit reversible transformation to CoTPP at various potentials upon alkaline conditions but this phenomenon did not exist in acidic and neutral catalytic environments.

CoTPP is the simplest structure of cobalt porphyrins, and the modification of CoTPP could start with the introduction of strong polar groups. On one hand, strong polar groups can enhance the interactions between porphyrins and conductive substrates; on the other hand, strong polar groups might improve the stability of intermediates. In 2016, Wu et al. loaded a carboxyl-functionalized cobalt porphyrin (CoTCPP in Figure 3) onto a BiVO_4_-containing electrode for photoelectrochemical OER upon neutral conditions [49]. Considering the presence of carboxyl groups, CoTCPP was generally believed to be unsuitable for heterogeneous electrocatalysis under alkaline conditions. In 2020, Villagrán et al. intercalated CoTCPP into zirconium phosphate (ZrP) bearing a layered structure as a molecular electrocatalyst (CoTCPP/ZrP) for OER [50]. Under a basic environment (0.1 M KOH), the carboxyl-functionalized porphyrin existed in the form of disodium salt but it was tightly embedded between ZrP layers. The intercalated species still showed positive heterogeneously electrocatalytic activity and remarkable catalytic stability.

In 2020, a single pyridyl porphyrin (CoP-py in Figure 3) was adsorbed on the surface of mesoporous TiO_2_ for electrocatalytic water splitting by Ozawa et al. [51]. When used as a catalyst, the individual FTO/TiO_2_/CoP-py anode did not behave well with a higher overpotential (*η*_10_ = 540 mV). Nevertheless, when CoP-py and PtP-py served as anode catalyst and cathode catalyst (Figure 4), respectively, the device showed wonderful comprehensive performance that could simultaneously evolve H_2_ and O_2_ in a molar ratio of 2:1, with nearly quantitative Faradaic efficiencies. Importantly, the electrocatalysis was performed under weak alkaline conditions, which was, in principle, disadvantageous for H_2_ evolution reaction. Therefore, this is a rare catalytic system of porphyrin-based compounds, which can synergistically improve the two half-reactions involved in water splitting.

Interestingly, Das et al. developed another alternative method that could activate generally inert materials for electrochemical OER [52]. In their study, CoTMPP (for structure, see Figure 3) was encapsulated inside a well-known MOF zeolitic imidazolate framework-8 (ZIF-8), both of which were inactive to water oxidation when used independently. However, the synergistic interactions between CoTMPP and ZIF-8 could change the chemical microenvironment after assembly and the CoTMPP@ZIF-8 electrocatalyst revealed heterogeneous OER activity with favorable robustness and efficiency at a wide pH range (pH = 2~7). The corresponding electrocatalytic performance and stability could be compared to cobalt porphyrins containing electron-withdrawing groups (Table 1). This study provides an excellent reference, not only to activate a system from an inert environment to an active one, but also to stabilize it using the encapsulation method.

**Figure 3 ijms-23-06036-f003:**
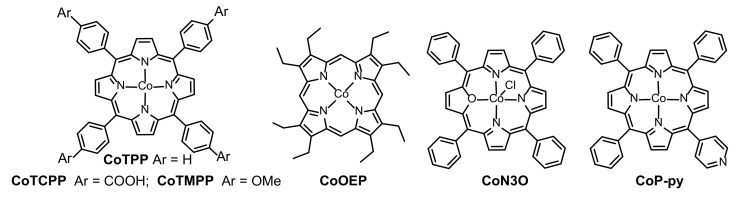
Chemical structures of noncovalently surface-immobilized porphyrin-based small molecules for heterogeneously electrochemical OER. Reproduced from ref. [47,49,51,52].

**Figure 4 ijms-23-06036-f004:**
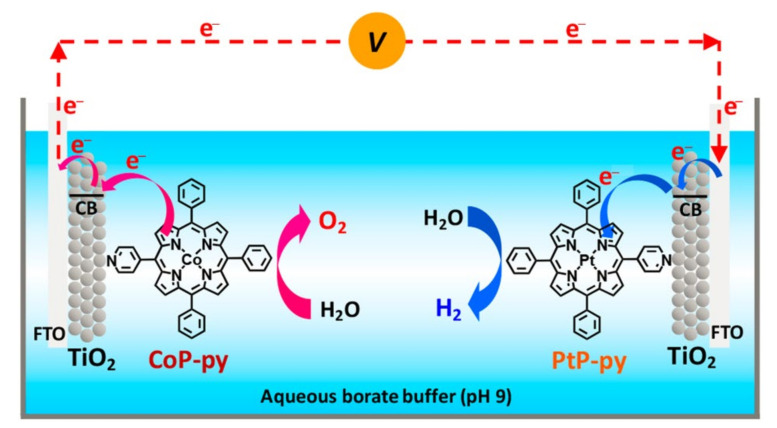
Schematic diagram of electrochemical cell for water splitting with two TiO_2_ electrodes containing CoP-py and PtP-py as molecular catalysts. Reprinted with permission from ref. [51]. Copyright 2020 American Chemical Society.

For the noncovalently immobilized strategy, the catalyst desorption might cause a decrease in catalytic performance, which is unsurprising since the noncovalent interactions are affected by many factors, such as the molecular polarity and stability of metalloporphyrins, surface modification of conductive substrate, counterions in electrolytes, and pH values of solutions [53,54]. Moreover, when the adsorption amount exceeds a certain limit, the catalytic activity of metalloporphyrins at a distance from the conductive supports is usually low, leading to the inefficient utilization of catalysts. In order to avoid catalyst desorption and to improve the utilization efficiency of catalysts as much as possible, fixing metalloporphyrins on the surface of conductive substrates using the covalent linking method is an alternative solution.

As mentioned above, Mn porphyrin dimers were initially investigated as homogeneous catalysts, and only porphyrins close to the electrode could provide catalytic roles. Hence, four Mn porphyrin dimers (Mn_2_DP-a~d) containing different numbers of phosphonic acid motifs were covalently grafted onto the surface of an ITO electrode (Figure 5A) [55]. Though the saturation coverages of Mn_2_DP-a~d were relatively low (4.3~7.8 × 10^−12^ mol·cm^2^), the assemblies exhibited excellent electrocatalytic performances with low overpotentials (*η*_10_ = 260~560 mV), high turnover numbers (TONs, 4.2~8.5 × 10^5^), and high TOFs (23.4~47.4 s^−1^) in various solutions (acidic, neutral, and basic). By comparing Mn_2_DP-a|ITO, Mn_2_DP-b|ITO, and Mn_2_DP-c|ITO, it was proved that the porphyrin with electron-withdrawing substituents at the *meso*-position behaved better than those with the electron-donating groups. In conducted, long-term electrolysis experiments (11 h), the catalysts (Mn_2_DP-a~c) bearing the monophosphonic acid group exhibited catalyst detachment behaviors. On the contrary, Mn_2_DP-d|ITO containing six extra phosphonic acids that could strongly bind to the ITO electrode displayed continuous oxygen evolution for 23 h with no desorption detected.

By virtue of the covalent surface immobilization method, Cao et al. systematically studied the effect of central atoms (such as Mn, Fe, Co, Ni, and Cu) in porphyrins on electrocatalytic performances [56]. As illustrated in Figure 5B, metalloporphyrins (MnP, FeP, CoP, NiP, and CuP) were covalently immobilized onto the surface of carbon nanotubes (CNTs) via 1,2,3-triazole linker. Metal ions had the same coordination environment and metalloporphyrins were independently attached to CNTs; therefore, the aggregation effect could be excluded as far as possible. All the metalloporphyrins@CNT simultaneously showed OER and HER activities. Under the same electrocatalytic conditions, CoP@CNT revealed better OER performance and FeP@CNT exhibited higher HER performance. Additionally, CoP@CNT and Fe@CNT were utilized as active catalysts to be assembled onto the electrodes of a water electrolysis cell and the catalytic performance was comparable to that of noble metal catalysts.

### 2.2. Porphyrin Coordination Polymers for Electrochemical OER

In the noncovalent or covalent immobilization strategies, the active catalytic sites are generally located near the surface of electrodes, so the limited electrode areas impose restrictions on the improvement of catalytic performances. Assembling porphyrins into polymers has been demonstrated to effectively achieve heterogeneous catalysis [19]. In particular, 2D or 3D polymers are meaningful to drastically enhance the density of active sites, resulting in the improvement of catalytic performances [57,58]. Gratifyingly, the *meso*-positions of porphyrin could be easily derivatized (such as pyridine, hydroxyl, and carboxylic acid), which could be used to construct amorphous or crystallizable porphyrin coordination polymers for OER.

For porphyrin-based small molecules, cobalt porphyrins have been proved to show better electrochemical OER performances than those of other metalloporphyrins. However, with the introduction of a second coordination metal ion, the interactions between metal ions would have considerable effects on the overall performances. In 2016, Grumelli et al. systematically studied the effect of different metal ions with the aid of several amorphous bimetallic coordination catalysts (M^1^TPyP-M^2^ in Figure 6) [59]. A series of monolayer coordination polymers were synthesized using subliming M^1^TPyP monomer (M^1^ = Fe or Co) onto the surface of Au (111), followed by evaporating a second metal ion (M^2^ = Fe or Co) via an electron beam evaporator. Both the monomers and coordination polymers were conducted for electrochemical OER and their performances approximately obeyed the following order: M^1^TPyP < M^1^TPyP-Fe < M^1^TPyP-Co. It could be found that the catalytic performances of coordination polymers were better than those of monomers, confirming the superiority of polymer strategy. The catalytic performances of M^1^TPyP-Co (that is M^2^ = Co) were better than those of M^1^TPyP-Fe. Interestingly, CoTPyP-Fe was observed to behave better than FeTPyP-Fe while FeTPyP-Co was discovered to be better than CoTPyP-Co, implying the synergistic effect between two metal centers. The extended conjugated structure of porphyrin macrocycle together with the surface *trans* effect were supposed to reasonably explain the synergistic effect.

In 2020, Wang and Zhu et al., combined porphyrin and porous titanium coordination polymer (TiPCP) to construct a push-pull type porous coordination polymer (TiCP-PCP in Figure 7A) [60]. Actually, porphyrin and TiPCP had been independently proved to have electrocatalytic OER activities. Fortunately, the combination of two components brought about a positive synergistic effect. The interactions between porphyrin subunits and Ti complex subunits facilitated the charge transfer and porous structures provided more active sites, leading to improved electrocatalytic performances (for detailed data, see Table 1). Subsequently, TiCP-PCP was further served as loading matrix to support another efficient electrocatalytic catalyst cobalt sulfide [61]. Upon experimental optimizations, the composite catalyst 0.2TiCP-PCP@Co_1−x_S (the mass percentage of TiCP-PCP was 20%) was observed to show better electrocatalytic performance. As illustrated in Figure 7B, the synergistic effect of multiple positive factors promoted the improvement of catalytic performance, including high active surface area, more efficient OER kinetics, larger Co^3+^/Co^2+^ ratio, enhanced adsorption capability of intermediates, and proper interfacial interactions. These works provide excellent examples for researchers to reasonably design efficient electrocatalysts by combining several positive factors into one system.

As a special type of coordination polymer, metal–organic frameworks (MOFs) possess many advantages, such as periodically porous structures, high specific surface areas, high flexibility, and favorable controllability [62,63]. The porphyrins as well as the coordinated metal ions or ion clusters could act as potentially catalytic active sites. Therefore, porphyrin-based MOFs are also suitable candidate catalysts for electrochemical OER.

Although porphyrin-based MOFs have been reported very early, only in recent years have they started to attract increasing attention for electrocatalytic OER [18]. In 2016, Dai and Sun et al. reported a lead–porphyrin MOF (Pb-TCPP in Figure 8) that exhibited electrocatalytic activity for OER [64]. In the MOF, lead ions acted as metal centers that coordinated with carboxyl functional groups of porphyrins and it was supposed that the porphyrin rings were the catalytic active centers after the center proton atoms were seized in an alkaline environment. Although the catalytic performance was not ideal, this study provided solid evidence for a porphyrin-based MOF that could serve as an electrocatalyst for OER. Shortly after this work, Morris et al. demonstrated a novel approach for cooperative catalysis. In their study, Zr–oxo clusters were bridged by Ni–porphyrin linkers in MOF PCN-224-Ni (for structure, see Figure 8) [65]. The large surface area (2600 m^2^·g^−1^) and the high pore volume (1.56 cm^3^·g^−1^) greatly enhanced the density of catalytic sites. As shown in Figure 9, with the nickel atoms serving as catalytic active centers, the electrochemical OER was supposed to undergo the following processes: the porphyrin ring was first oxidized to Por^+•^-Ni(II), followed by the binding of an H_2_O molecule to Ni(II) atom to provide Por^+•^-Ni(II)-OH_2_, and after consecutive proton transfer, electron transfer, and proton-coupled electron transfer processes, the O_2_ molecule was evolved from Por–Ni(II)–OO^•^ and the neutral Por–Ni(II) complex was regenerated to start a new catalytic cycle. In addition, the Zr–oxo nodes were assumed to act as proton acceptors via a cooperative approach. Considering one Zr–oxo node could only accept six protons and every Ni–porphyrin linker would release four protons per catalytic cycle, this reasonably explained the low TON (0.72) of electrochemical performance.

The application of simple porphyrin-based MOFs as electrocatalysts for OER is in its infancy and the undesirable catalytic results indicate that there is still much room for improvement. Given the large specific surface areas of MOF materials, the porphyrin-based MOFs could be utilized as supports to load other components to enhance catalytic performances. In 2017, Gu and Zhang et al. successfully applied this strategy to construct an efficient composite catalyst for electrochemical OER [66]. In the process of preparing catalysts, an epitaxial encapsulation approach was adopted. As represented in Figure 10, a precursor Ce(pdc)_3_ was first deposited onto PIZA-1 MOF (for chemical structure of PIZA-1, see Figure 8) to form Ce(pdc)_3_@PIZA-1 and a monodisperse CeO_2_@PIZA-1-400 thin film catalyst would be gained after calcining Ce(pdc)_3_@PIZA-1 at 400 °C. The resultant CeO_2_@PIZA-1-400/FTO catalyst revealed lower electrocatalytic overpotential (*η*_10_ = 370 mV) and faster charge transfer dynamics (Tafel slope 47.6 mV·dec^−1^) under 1.0 M KOH conditions, which was superior to those of PIZA-1-400/FTO and most other catalysts. The function of the monodisperse CeO_2_ nanoparticles was supposed to create more oxygen vacancies to enhance the electrochemical performance. Shortly afterward, a more abundant, low-cost, and non-toxic Fe(Salen) was encapsulated into the pores of PIZA-1 via a similar method by the same researchers [67]. It was observed that the Fe^3+^ sites in the composite catalyst were helpful to stabilize the −OOH intermediate species in electrocatalytic process, leading to the improved electrocatalytic activity.

### 2.3. Porphyrin-Based Porous Organic Polymers for Electrochemical OER

In addition to coordination polymers, porous organic polymers (POPs) are important porous materials, the structural units of which are mainly connected by light elements, such as carbon, nitrogen, and oxygen. Therefore, they are provided with the advantages of high specific surface area and high chemical stability, as well as low density, and have been widely applied in energy, environment, and catalysis fields [68,69]. Additionally, porphyrin derivatives are important building blocks for POPs and porphyrin-based POPs have been used as electrocatalysts [70].

Conjugated mesoporous polymers or conjugated microporous polymers (only the pore sizes are different and both are abbreviated as CMPs) are a class of amorphous POPs with extended π-conjugated lengths. Owing to their perfect electron transport features, they could be used as porous polymeric carbon materials to support metal catalysts to enhance catalytic activities [71,72]. In 2016, by virtue of the efficient Suzuki coupling reaction followed by pyrolysis (800 °C), Du et al. prepared a series of cobalt porphyrin-based CMPs (CoP-nph-CMP in Figure 11) for electrochemical OER [73]. They were all found to contain both micropores as well as mesopores and simultaneously exhibit electrocatalytic activities for OER and HER. CoP-2ph-CMP-800 was observed to behave better with lower overpotential and a lower Tafel slope than those of CoP-3ph-CMP-800 and CoP-4ph-CMP-800, which could be attributed to the high content of cobalt and high graphitization of carbon skeleton. Moreover, CoP-2ph-CMP-800 was successfully assembled into an electrolytic cell as OER and HER catalysts for overall water splitting. Following this study, the cobalt central atom was replaced by copper atom and the resultant Cu-CMP850 (for the structure of Cu-CMP precursor, see Figure 11) also showed OER and HER activities [74]. Although the performances of Cu-CMP850 were weaker than those of the corresponding cobalt porphyrin, they were much better than those of other copper complexes.

From the above-mentioned examples of porphyrin-based MOFs and POPs, for most cases, thermal annealing is needed to activate catalysts since it could induce the high graphitization of the molecular skeleton, which is helpful for improving the conductivity of materials. Nonetheless, heat annealing is a high energy consumption operation, meaning it is an expensive method. In 2019, Bhaumik, Pradhan, and Bhattacharya et al. cooperatively reported a CMP (Co-MMPy-1 in Figure 11) that exhibited positive electrochemical OER performance without any heat annealing [75]. It was supposed that Co-MMPy-1 was a donor-acceptor (D-A) microporous polymer, with cobalt porphyrin and pyrene serving as donor unit and acceptor unit, respectively. The D-A structure facilitated charge separation and the interactions between donor and acceptor promoted the kinetics of charge transfer. Therefore, the electrocatalytic OER performance of Co-MMPy-1 catalyst was comparable to those of catalysts with heat annealing (Table 1).

Recently, Yang and Hu et al. reported a series of porphyrin-based NiFe POP catalysts (M_1_TAPP-M_2_TCPP-POP in Figure 11) for electrochemical OER [76]. In their study, four POPs including FeTAPP-NiTCPP-POP (M_1_ = Fe, M_2_ = Ni), NiTAPP-FeTCPP-POP (M_1_ = Ni, M_2_ = Fe), FeTAPP-FeTCPP-POP (M_1_ = Fe, M_2_ = Fe), and NiTAPP-NiTCPP-POP (M_1_ = Ni, M_2_ = Ni) were prepared via amidation reactions, and both the monomers and POPs were tested for electrocatalytic OER. Firstly, POPs showed better performances than those of monomers, demonstrating the importance of polymeric structures. Secondly, the nickel-based monometallic POP (NiTAPP-NiTCPP-POP) exhibited superior OER activity than that of FeTAPP-FeTCPP-POP. Finally, both bimetallic POPs were found to behave better than those of monometallic POPs, while FeTAPP-NiTCPP-POP revealed the best activity with lower overpotential (*η*_10_ = 338 mV) and a lower Tafel slope (52 mV·dec^−1^), the performance of which was even better than most other 2D polymers (Table 1). The enhanced performance, to some extent, could be ascribed to the synergistic interactions between Ni and Fe atoms. XPS measurements and theoretical simulations indicated that the interactions enhanced the electron densities of Ni atoms in FeTAPP-NiTCPP-POP, making it easier to form oxygenated intermediates and improve the electrochemical OER activity.

Covalent organic frameworks (COFs), as a special class of POPs, have attracted tremendous attention for their periodically and uniformly porous structures [77,78]. Porphyrin rings have favorable catalytic properties and the macrocyclic centers can coordinate with various metal ions so as to provide encouraging catalytic potentials for porphyrin-based COFs [58,79,80]. Recently, reports of porphyrin-based COFs for OER applications have emerged. In 2020, Wang, Zhao, and Zhu et al. reported a classic Schiff-base type 2D covalent organic polymer (CoCOP in Figure 12) based on cobalt porphyrin [81]. The larger specific surface area (289 m^2^/g) and lower charge transfer resistance (R_ct_) rendered faster reaction kinetics. The periodically porous structure facilitated the diffusion of ions and the escape of gas products during electrolysis. Additionally, the structure of porphyrin and Schiff-base was also a D-A architecture that was favorable for charge separation and charge transfer processes. In the light of the above-mentioned factors, CoCOP generated positive overall electrochemical OER activities. Similar to this report, Gu et al. designed a series of 3D porphyrin-based COFs for electrocatalytic OER [82]. Under optimal conditions, PCOF-1-Co (for structure, see Figure 12) was demonstrated to perform better than other counterparts.

Graphdiynes (GDYs) are a novel class of carbon materials that are composed of aromatic rings connected by two acetylenic linkages. With unique hybrid π-conjugated networks, GDYs are believed to have semiconductor features with high carrier mobility, making them suitable candidates to be electrocatalytic nanomaterials [83,84,85]. However, due to the lack of efficient active centers in normal aromatic rings, they are merely used as support materials. Consequently, porphyrin rings bearing high catalytic activities were introduced into GDYs. In 2019, Chen and Zhang et al. reported a porphyrin-based COF nanosheet (Co-PDY in Figure 13) as catalyst for electrochemical OER [86]. In their study, the polymer was prepared using coordination, followed by polymerization onto the surface of copper foam (CF). The favorable conductivity of GDY, the porous structure of COF, and the catalytic active centers of porphyrin rings contributed to a high performance for OER, the overpotential of which was as low as 270 mV. Moreover, Co-PDY was also demonstrated to behave with high electrochemical HER activity. Shortly after this report, Zhao and Ding et al. employed a strategy of polymerization followed by coordination to construct the crystallizable porphyrin-based GDY and also achieved electrocatalysis for OER and HER [87]. Notably, in addition to planar GDY COFs, curved GDY COF was also successfully proved for electrochemical OER by Du et al. [88]. (COP)_n_-MWCNTs (Figure 13) was synthesized onto the surface of multiwalled carbon nanotubes (MWCNTs) that served as template and support. The Tafel slope of (COP)_n_-MWCNT catalyst for electrocatalytic OER was reduced to 60.8 mV·dec^−1^.
ijms-23-06036-t001_Table 1Table 1Summarized performances of porphyrin-based materials for electrochemical OER.CatalystsWorking ElectrodeElectrolyteLoading*η*_10_ (mV)Tafel Slope (mV·dec^−1^)RefCuTMPyPFTO0.10 M phosphate buffer (pH 7.0)1.0 mM310 ^a^—[41]CoTCPP/ZrPGC0.1 M KOH3.38 wt%47676.4[50]CoP-py/TiO_2_FTO0.1 M borate buffer (pH 9.0)0.10 μmol/cm^2^540—[51]CoTMPP@ZIF-8GC0.1 M KCl (pH 7.0)one CoTMPP per 15 cages387.4 ^b^210.3[52]Mn_2_DP-a|ITOITO25 mM Na_2_B_4_O_7_/0.1 M NaClO_4_ (pH 7.0)6.8 × 10^−12^ mol/cm^2^47090[55]CoP@CNTGC0.1 M KOH5.25 μg/mg ^c^422—[56]FeTPyP-CoAu(111)0.1 M NaOH—310—[59]TiCP-PCPCFP1.0 M KOH0.33 g/cm^2^310117[60]0.2TiCP-PCP@Co_1−x_SCFP1.0 M KOH0.33 g/cm^2^15765[61]Pb-TCPPGC1.0 M KOH—470106.2[64]PCN-224-NiFTO0.1 M NaClO_4_ (pH 7.0)8 × 10^13^ sites/cm^2^450150[65]CeO_2_@PIZA-1-400FTO1.0 M KOH—37047.6[66]Fe(Salen)@PIZA-1-400FTO1.0 M KOH—34056[67]CoP-2ph-CMP-800GC1.0 M KOH—37086[73]Cu-CMP850GC1.0 M KOH0.28 mg/cm^2^350135[74]Co-MPPy-1GC1.0 M NaOH4.23 × 10^−9^ mol/cm^2^42058[75]FeTAPP-NiTCPP-POPGC1.0 M KOH—33852[76]CoCOPCFP1.0 M KOH—350151[81]PCOF-1-CoGC1.0 M KOH—38689[82]Co-PDYCF1.0 M KOH—27099[86](CoP)n-MWCNTsGC1.0 M KOH0.14 mg/cm^2^43060.8[88]^a^ The value was recorded as onset potential. ^b^ The current corresponded to the overpotential of 1 mA/cm^−2^. ^c^ The value corresponded to the content of cobalt metal in CoP@CNTs determined by ICP-MS.

For comparison, the electrocatalytic OER performances of porphyrin-based materials for water oxidation are listed in Table 1. Generally speaking, the performances of porous polymers are better than those of small molecular systems. 0.2TiCP-PCP@Co_1−x_S showed the lowest overpotential of 157 mV while FeTAPP-NiTCPP-POP exhibited the smallest Tafel slope of 52 mV·dec^−1^, both of which contain at least two kinds of metal ions, indicating the key role of the synergistic effect between different metal ions. The performances of porphyrin-based MOFs serving as support materials have superior catalytic activities in comparison to pure frameworks owing to the improved electron transfer efficiency. Except for several examples, the overpotentials (*η*_10_) and the Tafel slope are difficult to balance, indicating a big challenge in simultaneously improving thermodynamics and kinetics. Therefore, further optimizations are required to enhance the overall performances.

## 3. Porphyrin-Based Systems as Bifunctional Oxygen Electrocatalysts for Zn–Air Batteries

As discussed above, the catalysts are mostly needed to catalyze the OER process when they are applied for water oxidation. Nonetheless, when they are used as catalysts for rechargeable zinc–air batteries, the catalysts are required to catalyze both the OER and ORR bidirectional processes [89]. A classic zinc–air battery is composed of a zinc anode as well as an oxygen-permeable cathode assembled in alkaline electrolyte, which possesses the advantages of low cost, safety, and high theoretical specific energy [3]. Catalysts are generally fixed on the cathodes. Due to the excellent catalytic potentials of porphyrin-based systems, both small molecules and porous polymers have been widely investigated as bifunctional oxygen electrocatalysts for zinc–air batteries.

### 3.1. Porphyrin-Based Small Molecules as Bifunctional Oxygen Electrocatalysts

According to the examples discussed in Section 2 and Section 3, it could be concluded that the metal center and the *meso*-substituent play pivotal roles in determining the electron density of metal centers in metalloporphyrins. Hence, the fine-tuning of metal center and the *meso*-substituent structure is an appealing approach to regulating the catalytic activity. Significantly, the *meso*-substituents of porphyrin-based catalysts for water splitting are mainly electron-withdrawing substituents. Taking the bifunctional oxygen electrocatalysis into consideration, both electron-donating and electron-withdrawing substituents are strategically investigated.

CoTPP was noncovalently immobilized onto the surface of MWCNT via a pyrene–pyridine (Py-Py) linker [90] and the electrocatalytic activities of CoTPP/MWCNT/Py–Py for OER and ORR were better than those of CoTPP/MWCNT, indicating the positive effect of Py-Py linker. Cao and Wang et al. compared the catalytic performances of CoTPP and Co-P (for structure, see Figure 14) with electron-withdrawing substituents [91]. CoTPP/CNT was demonstrated to present superior performance for OER, ORR, and zinc–air batteries on account of its more favorable electronic structure and facilitated charge transfer rate. Cao and Liang et al. investigated the substituent position effect on the performances of bifunctional oxygen electrocatalysts [92]. 3,4,5-OMe-CoP showed better OER and ORR performances than 2,4,6-OMe-CoP (for structures, see Figure 14) and detailed experiments indicated that the enhanced performances could be attributed to the better charge transport ability, high mass transfer rate, and better hydrophilic property. 3,4,5-OMe-CoP was then assembled into a zinc–air battery as a cathode catalyst, which produced a moderate battery performance (Table 2).

Recently, Xu et al. provided another effective approach to regulating the catalytic activity of metalloporphyrin via modulating the molecular conformation [93]. In their study, monolayer Co-TMPyP (for structure, see Figure 14) was absorbed onto the surface of chemically converted graphene (CCG). As illustrated in Figure 15, the electrostatic interaction and π–π interaction between Co-TMPyP and CCG rendered the rotation of pyridinium motifs, resulting in the molecular flattening of Co-TMPyP. The flattening effect compressed the length of the Co–N bond in cobalt porphyrin and enhanced the charge density of cobalt atoms that served as catalytic active centers, leading to the enhanced charge transfer efficiency as well as the improved catalytic performance. Notably, the flattening effect was not observed when Co-TMPyP was loaded on reduced graphene oxide (rGO). The Co-TMPyP/CCG catalyst manifested high catalytic activities for OER, ORR, and HER, the performances of which were all superior to most molecular catalysts and it was the first reported tri-functional molecular catalyst. A Zn–air battery was also successfully assembled with Co-TMPyP/CCG as a cathode bifunctional catalyst and the battery showed excellent specific capacity (793 mA·h·g^−1^) and power density (225.4 mW·cm^−2^), both of which were also the highest values of molecular catalysts for Zn–air batteries.

More broadly, except for cobalt porphyrins, metalloporphyrins containing other metal atoms are worthy of further exploration to gain more insight into the structure−activity relationship. Si and Liu et al. designed a series of metalloporphyrins bearing Fe, Co, and Ni as central atoms as well as phenyl, thienyl, and 4-thiophenephenyl as *meso*-substituents [94]. The performances of electrochemical ORR and OER followed the sequence of CoTThP/C > CoTTP/C > CoTPP/C and CoTThP/C > NiTTP/C > FeTPP/C (for detailed structures, see Figure 14). Density functional theory (DFT) simulations indicated that the central atoms, the *meso*-substituents, and the interactions between them had great impacts on the bond length of M-N_por_, the charge density of the central atom, and the energy gap between LUMO and HOMO orbitals. The calculated change trends were generally consistent with the catalytic performances. Jiang et al. studied the effect of the immobilization mode of manganese porphyrin on the performances of bifunctional oxygen electrocatalysis (for covalent immobilization of MnTPP-cov/1, MnTPP-cov/5, and MnTPP-cov/10, see Figure 14, with 1, 5, and 10 referring to electrodeposition time) [95]. The ORR process showed high sensitivity to the immobilization mode while the covalently immobilized MnTPP-cov/5 exhibited the best performance, including higher catalytic current and catalyst selectivity toward H_2_O. Surprisingly, when conducted for OER measurements, the TOF of noncovalent MnTPP-noncov catalyst (36 h^−1^) was even higher than that of MnTPP-cov/5 (28 h^−1^), implying that the immobilization mode had less of an effect on the OER rate, which was mainly determined by the number of accessible catalysts in this study.

Based on the above-discussed examples, it could be speculated that iron porphyrins are believed to behave with moderate ORR and low OER activities. Inspired by the working mechanism of cytochrome c oxidases (CcOs), Cao et al. designed an enzyme-inspired iron porphyrin (FeEIP in Figure 14) bearing a tethered imidazole to circumvent this obstacle [96]. The tethered imidazole extended from the *meso*-substituent would bind with the iron atom of porphyrin, thus enabling it to protect the key intermediate Fe^V^=O from being attacked by OH^−^, leading to both enhanced ORR (*E*_1/2_ = 0.84 V) and OER (Tafel: 84 mV·dec^−1^) activities of the FeEIP/CNT catalyst. Successfully, FeEIP/CNT was assembled into a Zn–air battery as a cathode catalyst and the performances were comparable to that of the Pt/C-Ir/C composite catalyst.

In addition to simple metal ions, the synergistic effects between distinctive metal ions are also studied. Yao and Tang et al. grew Co–/Ni–porphyrin complexes (Co^2+^/THPP and Co^2+^/THPP in Figure 14) onto the surface of rGO via a layer-by-layer (LBL) technique. The resulting rGO/(Ni^2+^/THPP/Co^2+^/THPP)_n_ (n refers to the cycle number) exhibited both OER and ORR activities [97]. Unsurprisingly, both the catalytic OER and ORR activities increased first and then decreased with the increasing of n value, implying that a balance was required between catalytic active centers and diffusion obstacles. Both the highest performances of OER and ORR corresponded to the rGO/(Ni^2+^/THPP/Co^2+^/THPP)_8_ catalyst and the Tafel slope of OER was as low as 50.17 mV·dec^−1^. Interestingly, the synergistic effect between different metal ions only manifested in the catalytic OER process, the reasons for which were needed to be further explored. Recently, Cao and Zhang et al. investigated the composite catalyst of (Co–P)_x_(Fe–P)_y_@CNT (for structures of Co–P and Fe–P, see Figure 14) for ORR, OER, and Zn–air batteries [98]. (Co–P)_0.5_(Fe–P)_0.5_@CNT was recorded to show better performances than those of Co–P@CNT, Fe–P@CNT, (Co–P)_0.7_(Fe–P)_0.3_@CNT, and (Co–P)_0.3_(Fe–P)_0.7_@CNT. These examples provide encouraging references to regulate the catalytic activities of catalysts by modulating the charge distributions of catalytic active centers through a variation of the interactions between different metal atoms.

### 3.2. Porphyrin-Based Porous Polymers as Bifunctional Oxygen Electrocatalysts

In terms of the fact that porous polymers are provided with more catalytic active sites and facilitated charge transfer routes, porphyrin-based porous polymers are excellent candidates to be bifunctional oxygen electrocatalysts. Gutzler et al. probed the stability of FeTPyP-Co coordination networks (for structure, see Figure 6, M_1_= Fe, M_2_ = Co) for electrocatalytic OER and ORR, with the aid of scanning tunneling microscopy (STM) and X-ray absorption spectroscopy techniques to monitor the reaction processes [99]. Experimental results showed that the metalloporphyrin networks were unchanged during electrocatalytic ORR. On the contrary, when conducted as an OER electrocatalyst, the organic backbones would rapidly decompose with the formation of a new Co/Fe(oxy)hydroxide species as the real catalyst. Considering the availability of pores and the dispersity of active sites of MOFs, Dehghanpour et al. synthesized a PCN-224(Co)/MWCNT composite catalyst for electrochemical OER and ORR [100]. The preparation process of the catalyst did not require thermal annealing; however, the catalytic activities were not ideal, so the material design needed to be further optimized.

In light of the low performances of MOF catalysts, Huang and Zhou et al. designed a porphyritic zirconium MOF (PCN-226(Co) in Figure 16) to alleviate these restrictions [101]. The Zr(+4) cations could form strong coordination bonds with the carboxyl groups in porphyrins, leading to the enhanced chemical stability. In addition, the zirconium components in this MOF were Zr-oxide chains instead of Zr_6_ clusters in classic Zr-MOFs, and the smaller size of Zr-oxide chains rendered the packing density of functional groups much closer, resulting in more active sites and improved charge mobility by means of the hopping mechanism. Based on these factors, the PCN-226(Co)/C catalyst presented positive performances for both electrochemical OER and ORR (Table 2). A Zn–air battery was also assembled from PCN-226(Co)/C, the performances of which were comparable to those of Pt/C+RuO_2_ noble metal catalysts.

Shan and Kong et al. focused on porous covalent porphyrin frameworks (CPF, for framework structure, see TMPP in Figure 16) that could act as precursors for bifunctional oxygen electrocatalysts [102]. As outlined in Figure 17A, the CPF grafted with triphenylphosphine was first synthesized via Suzuki coupling reactions, and the resulting Fe-TPP-CPFs underwent thermal treatment to provide a Fe_2_P/Fe_4_N-T catalyst (T refers to pyrolysis temperature and 800 °C was the optimized temperature). Obviously, upon thermal annealing, the structures of triphenylphosphine, porphyrin rings, and framework would be greatly changed, leaving Fe_2_P and Fe_4_N loaded on N-doped carbon materials that could serve as catalytic active sites. Therefore, Fe_2_P/Fe_4_N-800 showed OER, ORR, and HER activities and the performances were comparable to other reported catalysts. Three years later, He et al. adopted another alternative approach to achieving the goals (Figure 17B) [103]. In their study, the porphyrins and crosslinking networks were simultaneously formed on the surface of CNTs, and the CNTs@(Fe,Co,Ni)PP-T catalysts were obtained after the following chelation with metal ions, as well as pyrolysis. Additionally, three metal atoms were incorporated together to modulate the charge density of catalytic active centers. The optimized CNTs@(Fe,Co,Ni)PP-800 catalyst exhibited high OER and ORR activities and was also utilized as a cathode catalyst to assemble a Zn–air battery with high performance (155.13 mW·cm^2^ at 0.645 V). With similar design strategy but different polymers, Wang and Lv et al. prepared a conjugated microporous polymer (CoPP-FePc-CMPs in Figure 16) based on cobalt porphyrin and iron phthalocyanine [104]. The polymer was pyrolyzed on the surface of graphene oxide, resulting in a Co, Fe, and N tri-doped graphene catalyst (CoFeNG). The CoFeNG was proved to show high electrocatalytic OER (*η*_10_ 360 mV, Tafel slope 68.9 mV·dec^−1^) and ORR (*E*_1/2_ 360 mV, Tafel slope 53.5 mV·dec^−1^) activities; however, the performance of the corresponding Zn–air battery was needed to be further optimized and improved (Table 2).

In pursuit of simple and efficient preparation of catalysts as well as to improve catalytic performances, Zhang et al. systematically investigated another series of porphyrin covalent organic frameworks (POF in Figure 16) [105]. The POFs could effectively be a one-pot templated synthesis from terephthalaldehyde and pyrrole on the surface of various supports, such as CNTs and reduced graphene oxide. The resultant composite materials could not only be used as catalysts themselves but also be applied as precursors or supports to prepare other catalysts. First, a thin layer POF was loaded on CNT scaffold to produce a catalyst (CNT@POF) for a Zn–air battery [106]. CNT@POF displayed excellent electrocatalytic performances including high specific capacity (772.7 mA·h·g^−1^), high peak power density (237 mW·cm^−2^), long-term stability, and low charge or discharge voltage gap (0.71 V). Notably, the low charge or discharge voltage gap was a symbol for excellent bifunctional ORR/OER activities. In addition, the CNT@POF thin film could even serve as a catalyst for a flexible all-solid-state Zn–air battery. In consideration of the perfect catalytic activities of POF, it was composited in situ with reduced graphene oxide to prepare G@POF-Co for OER and ORR electrocatalyst, the performances of which were comparable to IrO_2_ and Pt/C catalysts [107]. Subsequently, by pyrolysis of the G@POF-Co catalyst at 950 °C, the same researchers successfully obtained a new catalyst (Co-POC) that exhibited single-atom catalytic characteristics for OER, ORR, and a Zn–air battery [108]. The cobalt porphyrin macrocycles, the two-dimensional framework skeleton, and the hybridization with graphene were all critical factors that determined the formation and performance of a single-atom bifunctional oxygen electrocatalyst. In order to better balance the catalytic performances between OER and ORR, especially to improve electrochemical OER activity, the same researchers further adjusted the catalysts by means of composite with other inorganic nanoparticles such as Co_3_O_4_ [109] and NiFe-layered double hydroxides (LDH) [110]. The improved performances of Co_3_O_4_@POF and LDH-POF catalysts indeed confirmed their hypothesis.

As discussed previously, the pores in the porous materials including MOFs, conjugated microporous polymers, and COFs are usually microporous, and the limited aperture might restrict the effective mass transfer. Zhang et al. prepared a series of porphyrin-based imine gels, which possessed accessible active sites, hierarchical porosity, and high specific surface area to overcome this limitation [111]. Under condition optimization, the ferrocene-1,1′-dicarbaldehyde(NA)-based imine gels (MTAPP-NA in Figure 16) were demonstrated to display better overall performances. Under the same experimental conditions, the electrocatalytic OER activities of MTAPP-NA were CoTAPP-NA>NiTAPP-NA>> ZnTAPP-NA, PdTAPP-NA, and H_2_TAPP-NA, indicating the significant influence of metal catalytic active centers. Apart from OER, the CoTAPP-NA gel also showed electrocatalytic ORR and HER activities. Moreover, the tri-functional catalytic performances were all compared with the reported performances of other porous materials, confirming the success of this imine gel strategy.

**Figure 16 ijms-23-06036-f016:**
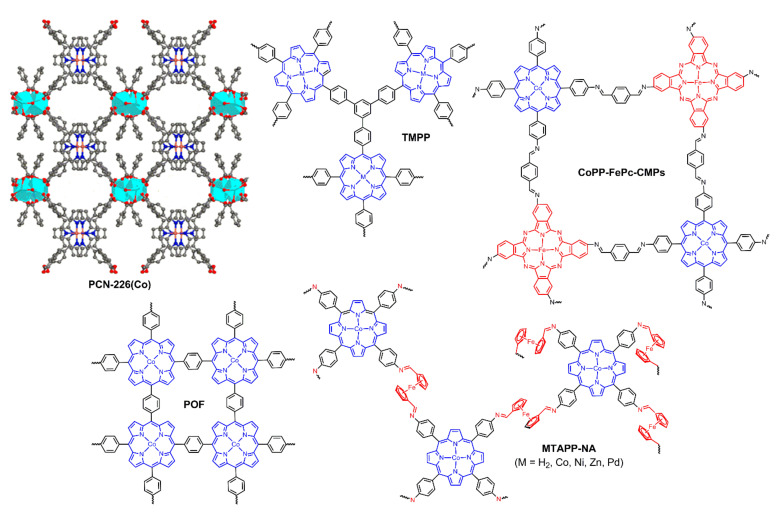
Porphyrin-based porous polymers for bifunctional oxygen electrocatalysis. Reprinted with permission from ref. [101]. Copyright 2020 American Chemical Society. Reproduced from ref. [102,104,105,111].

**Figure 17 ijms-23-06036-f017:**
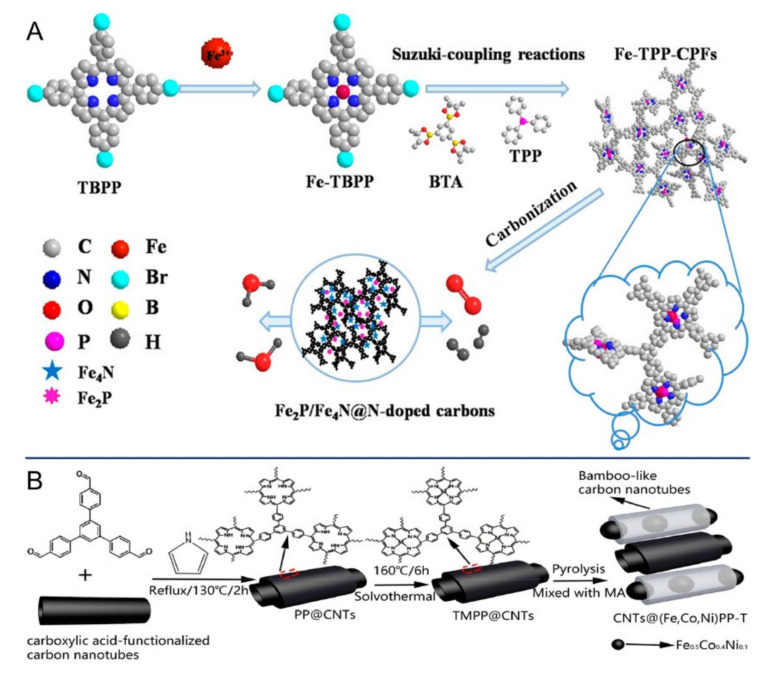
(**A**) Synthesis of Fe-porphyrin-based COFs and corresponding Fe_2_P/Fe_4_N@N-doped carbon electrocatalysts. Reprinted with permission from ref. [102]. Copyright 2017 American Chemical Society. (**B**) Preparation of CNTs@(Fe,Co,Ni)PP-T electrocatalyst. Reprinted with permission from ref. [103]. Copyright 2020 Elsevier Ltd. All rights reserved.

Both porphyrin-based small molecules and porous polymers have been demonstrated to reveal bifunctional oxygen electrocatalytic activities and most of them are also successfully assembled into Zn–air batteries as cathode catalysts. The performances of corresponding electrocatalysts are summarized in Table 2. The electrochemical ORR performances are overall superior to those of OER owing to the theoretically higher required potential for water oxidation. The performances of porous polymer-based catalysts are better than those of small molecules and porous organic polymer-based catalysts behave better than coordination polymers. The best reported peak power density of porphyrin-based catalytic systems is 237 mW·cm^2^, corresponding to the CNT@POF catalysts, and the CNTs@(Fe,Co,Ni)PP-800 catalyst possessed the lowest charge or discharge voltage gap of 0.69 V. To synergistically improve the performance of OER, ORR, and Zn–air batteries, there are still many issues to be explored.
ijms-23-06036-t002_Table 2Table 2Summarized performances of porphyrin-based materials for bifunctional oxygen electrocatalysts and Zn–air batteries.Catalysts
OER ^a^
ORR ^a^Zn–air Battery ^d^Ref.Electrolyte*η*_10_ (mV)Tafel Slope (mV·dec^−1^)ElectrolyteE_1/2_ (V)Tafel Slope (mV·dec^−1^)Voltage Gap ∆*E* (V)Peak Power Density (mW/cm^−2^)CoTPP/MWCNT/Py–Py1.0 M KOH440—0.5 M H_2_SO_4_0.375 ^b^———[90]CoTPP/CNTs1.0 M KOH40760.30.1 M KOH0.8136.9—155.7[91]Co-P/CNTs1.0 M KOH48071.60.1 M KOH0.7643.5—84.5[91]3,4,5-OMe-CoP/CNT1.0 M KOH482810.1 M KOH0.80490.8144.8[92]Co-TMPyP/CCG1.0 M KOH37962.40.1 M KOH0.82478.5—225.4[93]CoTThP/C0.1 M KOH5841100.1 M KOH0.74654——[94]FeEIP/CNT0.1 M KOH500840.1 M KOH0.84—0.89132.9[96]rGO/(Ni^2+^/THPP/Co^2+^/THPP)_8_1.0 M KOH340 50.170.1 M KOH0.84 ^c^———[97](Co–P)_0.5_(Fe–P)_0.5_@CNT0.1 M KOH420—0.1 M KOH0.8070.10.74174.5[98]PCN-224(Co)/MWCNTborate buffer (pH = 9.2)510 ^a^—0.5 M H_2_SO_4_—154——[100]PCN-226(Co)/C1.0 M KOH4451110.1 M KOH0.7558.9—133[101]Fe_2_P/Fe_4_N@C-8001.0 M KOH4101770.1 M KOH0.8065——[102]CNTs@(Fe,Co,Ni)PP-8001.0 M KOH35578.50.1 M KOH0.83763.850.69155.13[103]CoFeNG1.0 M KOH36068.90.1 M KOH0.77753.5—53.4[104]CNT@POF——————0.71237[106]G@POF-Co0.1 M KOH4301610.1 M KOH0.8146.9——[107]Co-POC0.1 M KOH4701390.1 M KOH0.8353.51.0478[108]Co_3_O_4_@POF0.1 M KOH390660.1 M KOH0.85381.00222.2[109]LDH-POF0.1 M KOH250990.1 M KOH0.80460.74185[110]CoTAPP-NA1.0 M KOH416681.0 M KOH0.84———[111]^a^ The current corresponding to this overpotential was 2 mA/cm^−2^. ^b^ The value was recorded as peak potentials. ^c^ The value was recorded as onset potentials. ^d^ The electrolyte for Zn–air battery was generally 6.0 M KOH with 0.2 M ZnAc_2_.

## 4. Conclusions and Perspectives

In summary, the bottlenecks of OER in kinetics put forward urgent requirements for high-performance catalysts. Owing to the easily derivatized structures and attractive optical and electrical properties, metalloporphyrins provide positive solutions for the design of high-performance electrocatalysts. Both porphyrin-based small molecules and porous polymers have been demonstrated as favorable catalytic systems for homogeneously or heterogeneously electrochemical OER. The catalyst utilization efficiency of small molecule homogeneous electrocatalysis is relatively low, so noncovalently and covalently immobilized strategies are developed to load the small molecule catalysts on conductive supports. However, many issues remain to be resolved using the surface-immobilized strategy. Porphyrin-based porous polymers, including both coordination polymers and porous organic polymers, have confirmed their great potential as electrocatalysts in view of their porous features, tunable structures, and large specific surface areas. The porous polymers could be used in various ways, such as directly serving as catalysts, as catalytic precursors, and as supports. Apart from electrochemical OER for water oxidation, porphyrin-based systems are also performed as bifunctional oxygen electrocatalysts. Surface-immobilized small molecules and porous polymers can also play important roles. Impressive achievements have been made, for example, 0.2TiCP-PCP@Co_1−x_S showed an overpotential as low as 157 mV, and Co-TMPyP/CCG catalyst exhibited excellent tri-functional catalytic activities for OER, ORR, and HER. By comparison, cobalt porphyrin showed superior performances than other metalloporphyrins, and the synergistic effect of introduction of other metals contributes to the further enhancement of catalytic activity. In addition to the material level, the catalytic mechanisms have also been deeply studied. Even with the same substrates, the intermediates would vary greatly upon different conditions; thus, these issues must be carefully considered in molecular design and condition optimizations. The above-mentioned factors provide solid references for the further design of more efficient electrocatalysts.

Although extraordinary progress has been made, the research on porphyrin-based catalysts for electrochemical OER are in the initial stages in comparison to ORR, HER, and CO_2_RR. In addition, the overall performances are still weaker than those of noble metal systems. Therefore, great efforts are in great demand. First of all, gaining deeper insight into the real catalytic active centers and the detailed catalytic mechanisms, including valence changes and charge transfer processes, is urgently needed. A clear structure–activity relationship is of great significance for further molecular design, and this might require more sophisticated structural designs or more advanced monitoring methods. Second of all, design and synthesis of more efficient porous polymer materials, whether as catalysts, catalyst precursors, or supporting materials, are attractive research topics. Simpler preparation methods, more active sites, larger specific surface area, smoother mass transport, and charge transfer channels would be very helpful to improve the overall performances. Subsequently, considering the adsorption and desorption of catalysts, charge injection and transfer on electrodes, and rapid diffusion of electrolyte and oxygen would also have considerable effects on the final performances; thus, the electrode preparation process and device structure are worth optimizing. Finally, although there is still a long way to go, commercialization of materials is one of the ultimate goals of research. In addition to improving electrocatalytic efficiency and stability, it is essential to carry out commercial research at the necessary stage.

## Figures and Tables

**Figure 1 ijms-23-06036-f001:**
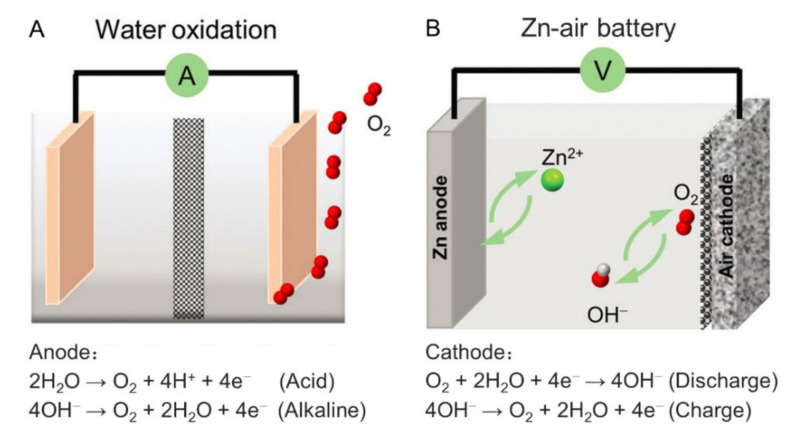
Schematic illustrations of representative devices for electrochemical OER: (**A**) water-splitting electrolytic cell and (**B**) rechargeable Zn–air battery. Reprinted with permission from ref. [28]. Copyright The Royal Society of Chemistry 2021.

**Figure 2 ijms-23-06036-f002:**
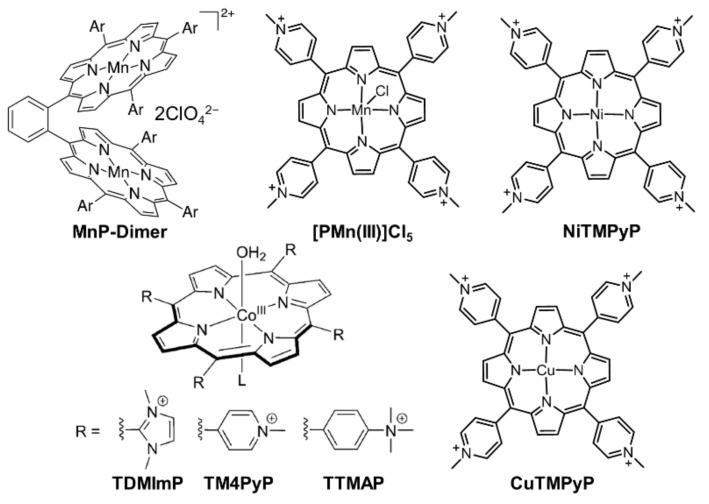
Chemical structures of porphyrin-based small molecules as homogeneous catalysts for electrocatalytic OER. Reproduced from refs. [37,38,39,40,41].

**Figure 5 ijms-23-06036-f005:**
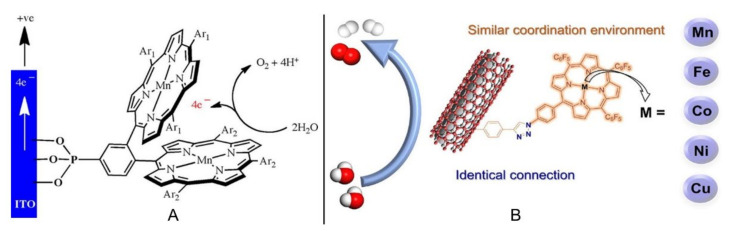
Covalently surface-immobilized porphyrins for electrochemical OER: (**A**) Mn porphyrin dimers (Mn_2_DP-a|ITO, Ar_1_ = Ar_2_ = C_6_F_5_; Mn_2_DP-b|ITO, Ar_1_ = Ar_2_ = C_6_H_2_(CH_3_)_3_; Mn_2_DP-c|ITO, Ar_1_ = C_6_F_5_, Ar_2_ = C_6_H_2_(CH_3_)_3_; Mn_2_DP-d|ITO, Ar_1_ = Ar_2_ = C_6_H_4_PO_3_H_2_) on ITO electrode. Reprinted with permission from ref. [55]. Copyright 2017 Elsevier Ltd. All rights reserved. (**B**) Metalloporphyrins covalently immobilized onto carbon nanotubes (abbreviated as MnP@CNT, FeP@CNT, CoP@CNT, NiP@CNT, and CuP@CNT). Reprinted with permission from ref. [56]. Copyright 2021 Elsevier Ltd. All rights reserved.

**Figure 6 ijms-23-06036-f006:**
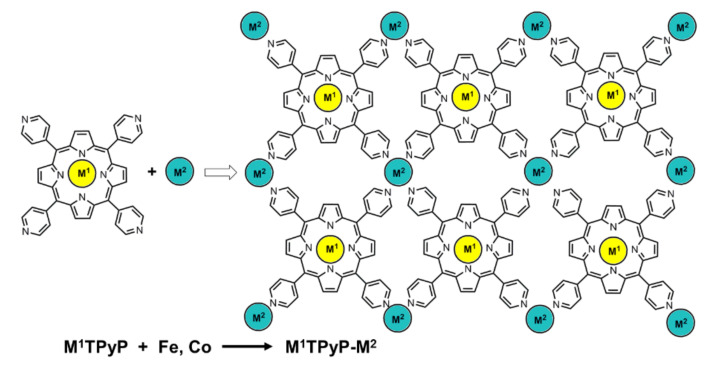
Synthesis and chemical structures of M^1^TPyP-M^2^. Reproduced from ref. [59].

**Figure 7 ijms-23-06036-f007:**
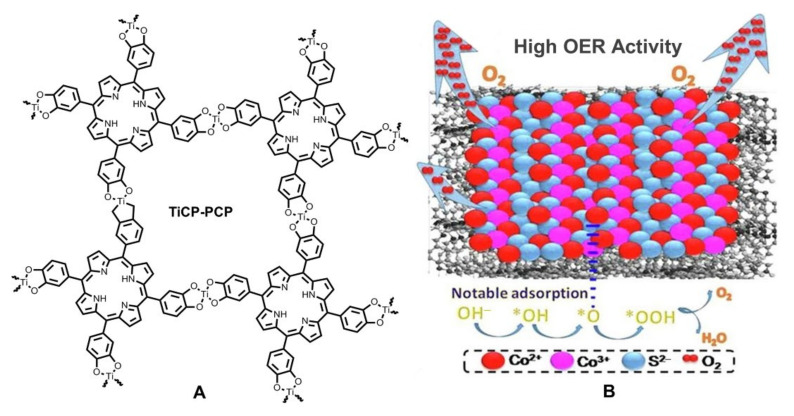
(**A**) Chemical structure of coordination polymer TiCP-PCP. Reproduced from ref. [60]. (**B**) Schematic representation of the catalytically active zone of 0.2TiCP-PCP@Co_1−x_S. Reprinted with permission from ref. [61]. Copyright 2020 Elsevier Ltd. All rights reserved.

**Figure 8 ijms-23-06036-f008:**
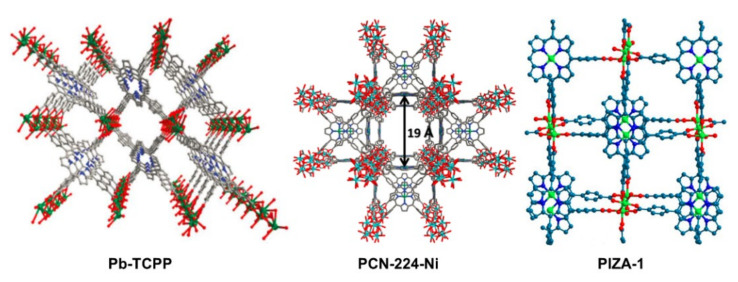
Porphyrin-based metal–organic frameworks for heterogeneously electrochemical OER. Reprinted with permission from ref. [64]. Copyright The Royal Society of Chemistry 2016. Reprinted with permission from ref. [65]. Copyright The Royal Society of Chemistry 2016. Reprinted with permission from ref. [66]. Copyright The Royal Society of Chemistry 2017.

**Figure 9 ijms-23-06036-f009:**
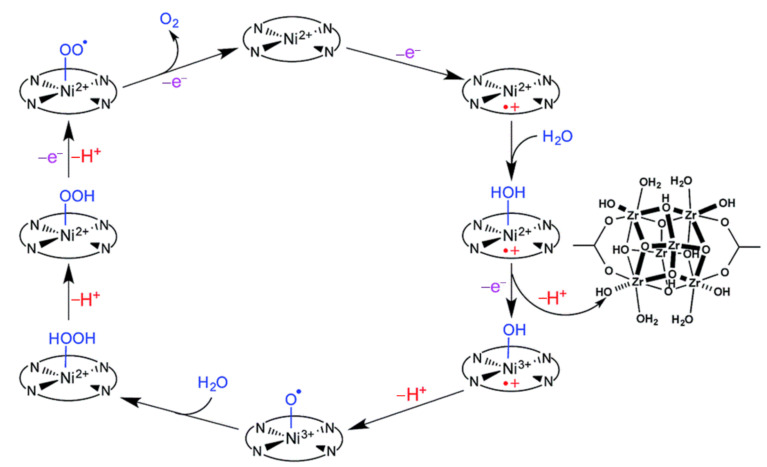
Proposed mechanism for the electrochemical OER catalyzed by PCN-224-Ni. Reprinted with permission from ref. [65]. Copyright The Royal Society of Chemistry 2016.

**Figure 10 ijms-23-06036-f010:**
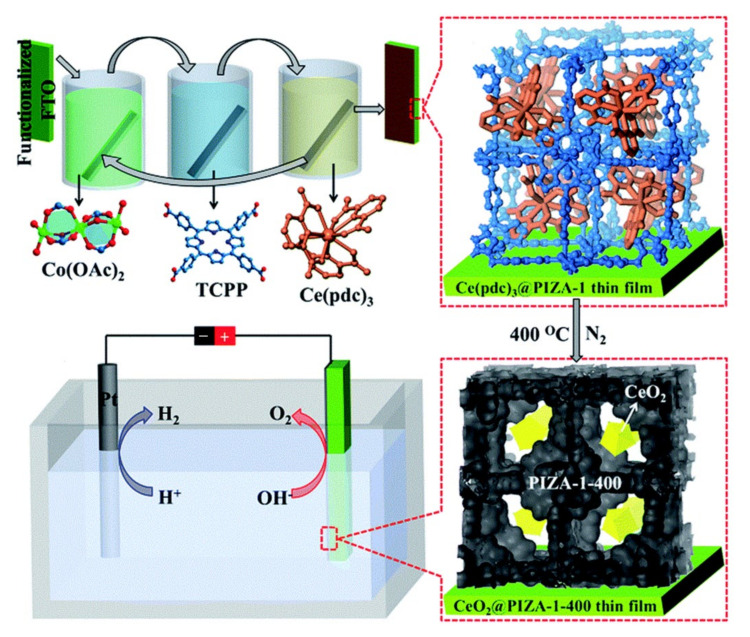
Schematic representation of the preparation process and electrocatalytic water splitting of CeO_2_@PIZA-1 thin film. Reprinted with permission from ref. [66]. Copyright The Royal Society of Chemistry 2017.

**Figure 11 ijms-23-06036-f011:**
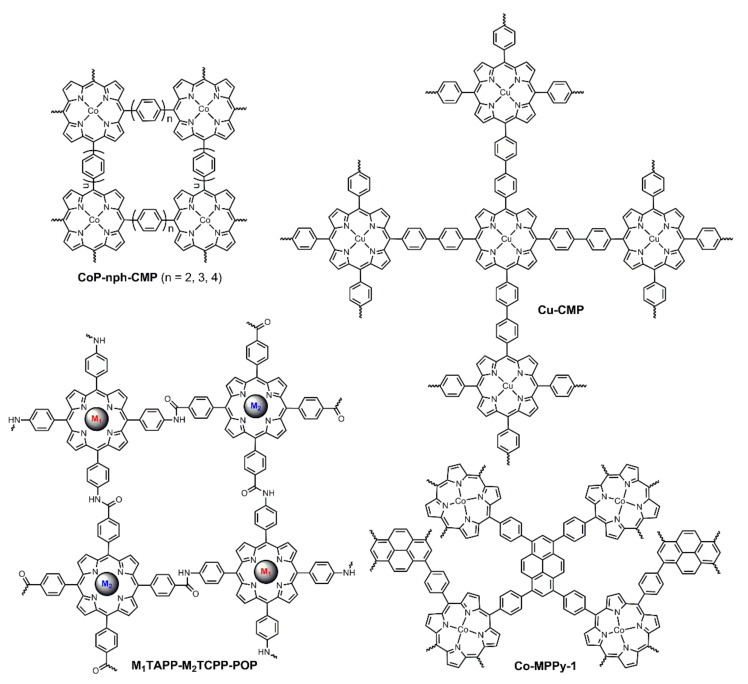
Porphyrin-based amorphous porous organic polymers for heterogeneously electrochemical OER. Reproduced from ref. [73,74,75,76].

**Figure 12 ijms-23-06036-f012:**
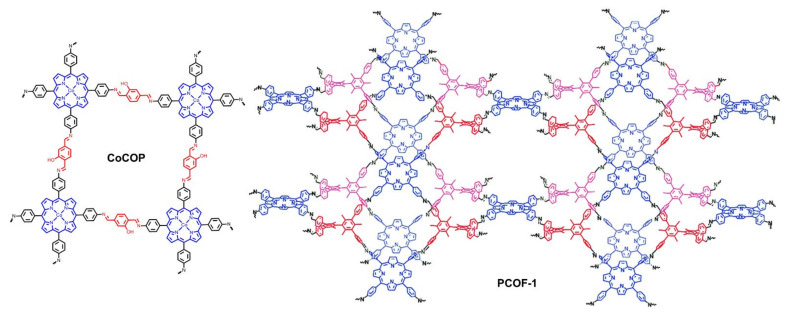
Porphyrin-based Schiff-base type covalent organic frameworks for heterogeneously electrochemical OER. Reproduced from ref. [81]. Reprinted with permission from ref. [82]. Copyright 2020 Elsevier Ltd. All rights reserved.

**Figure 13 ijms-23-06036-f013:**
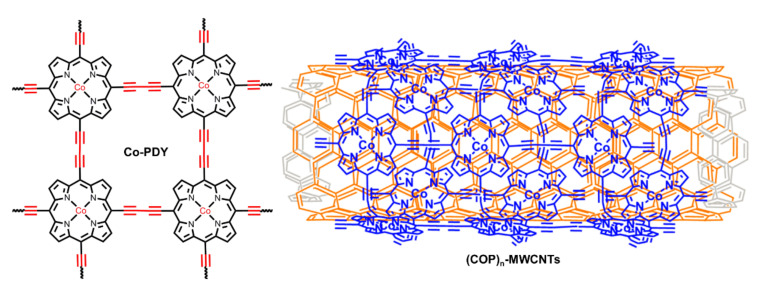
Porphyrin-based graphdiyne type covalent organic frameworks for heterogeneously electrochemical OER. Reproduced from ref. [86]. Reprinted with permission from ref. [88]. Copyright 2015 American Chemical Society.

**Figure 14 ijms-23-06036-f014:**
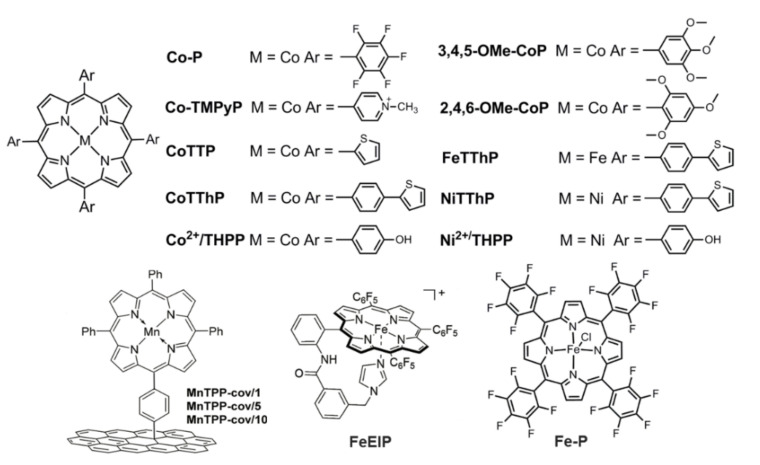
Porphyrin-based small molecules for bifunctional oxygen electrocatalysts. Reproduced from ref. [91,92,93,94,95,96,97,98].

**Figure 15 ijms-23-06036-f015:**
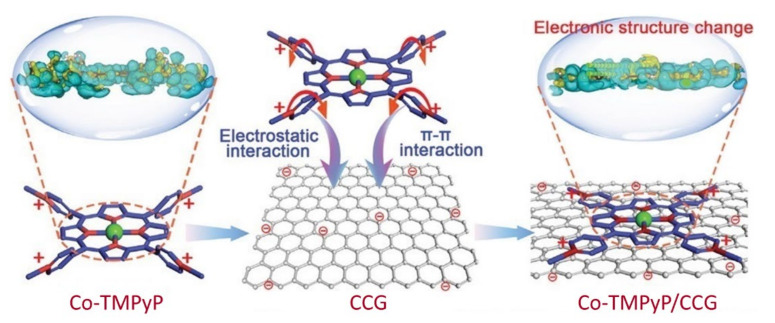
Schematic illustration of supramolecular assembly of Co-TMPyP on CCG. Reprinted with permission from ref. [93]. Copyright 2021 Wiley Online Library.

## Data Availability

Not applicable.

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
