# Peer review of "Recent Advances in Porphyrin-Based Systems for Electrochemical Oxygen Evolution Reaction†"

_ijms, 2022, doi:10.3390/ijms23116036_

Round 1

Reviewer 1 Report

The manuscript "Recent Advances in Porphyrin-Based Systems for Electrochemical Oxygen Evolution Reaction" has been submitted by Bin Yao, Youzhou He, Song Wang, Hongfei Sun and Xingyan Liu as a review to International Journal of Molecular Science. The review covers a relatively short period of time, from 2016 to 2021 in the title area. Authors describe porphyrin catalysts for electrochemical water oxidation and systems that serve as bifunctional oxygen electrocatalysts for Zn–air batteries. The porphyrins are divided into several categories: small molecules (for homogenous catalysis or immobilized on a solid surface), porphyrin coordination polymers and porous organic polymers. Over one hundred references are cited, and chosen physicochemical data are summarized in two tables. Such a review is for sure valuable, however, the manuscript is not very well written. The language lacks precision, there are several incomprehensible sentences, like „According to the results of section 2 and section 3, it could be discovered that the metal centers and the meso-substituents play pivotal roles in determining the electronic structures of metal centers in porphyrins, leading to the modulation of catalytic activities” or inaccurate expressions like „(...) lead atoms acted as coordination metals with carboxyl functional groups”, or „the H atoms being deprotonated...”. Also expressions, like „and so on”, should be avoided. The abbreviations of the compounds are inconsistent ([PMn(III)]Cl5 and NiTMPyP have analogous structures and totally different denotation). Authors may consider numbering of structures. In combination with uneasy language it makes reading relatively difficult. The graphics definitely needs improvement, as each artwork has a different style, different size and in many cases it is unreadable (e.g. Figure 12, Figure 15). Some of the graphics may be copied from original publications (by permission), however in such a case, it should be clearly stated in the figure caption. In my opinion, relatively simple graphics, like schemes of molecules, should be redrawn by Authors and fully unified (e.g. Figure 11 includes three different modes of showing an infinite structure – brackets with n, wavy line and short lines). Concluding, before accepting the work for publication, it should be corrected in terms of language, precision and graphics.

Author Response

For reviewer 1

The manuscript "Recent Advances in Porphyrin-Based Systems for Electrochemical Oxygen Evolution Reaction" has been submitted by Bin Yao, Youzhou He, Song Wang, Hongfei Sun and Xingyan Liu as a review to International Journal of Molecular Science. The review covers a relatively short period of time, from 2016 to 2021 in the title area. Authors describe porphyrin catalysts for electrochemical water oxidation and systems that serve as bifunctional oxygen electrocatalysts for Zn–air batteries. The porphyrins are divided into several categories: small molecules (for homogenous catalysis or immobilized on a solid surface), porphyrin coordination polymers and porous organic polymers. Over one hundred references are cited, and chosen physicochemical data are summarized in two tables. Such a review is for sure valuable, however, the manuscript is not very well written. The language lacks precision, there are several incomprehensible sentences, like „According to the results of section 2 and section 3, it could be discovered that the metal centers and the meso-substituents play pivotal roles in determining the electronic structures of metal centers in porphyrins, leading to the modulation of catalytic activities” or inaccurate expressions like „(...) lead atoms acted as coordination metals with carboxyl functional groups”, or „the H atoms being deprotonated...”. Also expressions, like „and so on”, should be avoided.

Reply: Thanks very much for sincerely pointing out these mistakes. The incomprehensible sentences, inaccurate expressions, and some non-essential vocabularies have been carefully corrected in the revised version. In view of the differences in language expression between eastern and western countries, the article has also been further proofread by two chemical researchers of native english speaker.

The abbreviations of the compounds are inconsistent ([PMn(III)]Cl5 and NiTMPyP have analogous structures and totally different denotation). Authors may consider numbering of structures. In combination with uneasy language it makes reading relatively difficult.

Reply: Thanks very much for the reviewer’s excellent suggestion. The abbreviations [PMn(III)]Cl5 and NiTMPyP are excerpted from reference 38 (Electrochim. Acta 2014, 135, 301–310) and reference 40 (Chem. Sci. 2019, 10, 2613–2622), respectively. Considering that they were published by different research groups with different writing habits, the abbreviations for compounds are greatly different. As a sign of respect for the original authors, most of the abbreviations in the manuscript are consistent with the original literatures. In the light of these abbreviations would also appear in other literatures, the forced unification of abbreviations is beneficial for this article to be read, but it would cause great confusion for researchers. Actually, with increasing researchers devote themselves to this field, some abbreviations have been generally accepted by researchers, such as 5,10,15,20-tetrakis(4-N-methylpyridinyl) porphyrin and 5,10,15,20-tetrakis(4-carboxyphenyl) porphyrin are abbreviated as TMPyP and TCPP, respectively. In fact, we also consider numbering of structures in the first draft of the manuscript. However, in terms of the great influences of central metals and meso-substituents on the performances of electrochemical oxygen evolution reaction, abbreviating the compounds is helpful for the readers to intuitively know the central metals and meso-substituents of metalloporphyrins at first sight. Therefore, we use abbreviations instead of numbering compounds.

The graphics definitely needs improvement, as each artwork has a different style, different size and in many cases it is unreadable (e.g. Figure 12, Figure 15).

Reply: Thanks very much for your excellent advice. The graphics have been reformatted in the in the revised version, which would be helpful for enhancing its readability.

Some of the graphics may be copied from original publications (by permission), however in such a case, it should be clearly stated in the figure caption.

Reply: Following the reviewer’s suggestion, the copied graphics have been clearly stated in the figure caption in the revised version.

In my opinion, relatively simple graphics, like schemes of molecules, should be redrawn by Authors and fully unified (e.g. Figure 11 includes three different modes of showing an infinite structure – brackets with n, wavy line and short lines). Concluding, before accepting the work for publication, it should be corrected in terms of language, precision and graphics.

Reply: Thanks very much for your nice advice. In fact, most of the structural formulas are redrawn since the graphs in the original literature are fuzzy. To respect originality, the expression of structural formula of polymers adopts the same form with the original literatures. In view of the expression style have little effect on the specific structures and experimental results, the expression style has been fully unified in the revised version. As mentioned above, the language and precision have also been corrected in the revised version.

Reviewer 2 Report

Title: Recent Advances in Porphyrin-Based Systems for Electrochemical Oxygen Evolution Reaction
##Overall Comments
This review article described metalloporphyrins which could be reasonable solutions for designing high-performance electrocatalysts. Porphyrin-based small molecules and porous polymers have demonstrated good catalytic systems for homogeneously or heterogeneously electrochemical OER. The paper writing is well with compact information. The reviewer would like to suggest including the following points:
(a)    Is there any metalloporphyrins that could be useful for commercial purpose? Please highlight
(b)    What would be regeneration approaches for metalloporphyrins?
(c)    What are research gaps for commercial use?

Author Response

For reviewer 2

This review article described metalloporphyrins which could be reasonable solutions for designing high-performance electrocatalysts. Porphyrin-based small molecules and porous polymers have demonstrated good catalytic systems for homogeneously or heterogeneously electrochemical OER. The paper writing is well with compact information. The reviewer would like to suggest including the following points:

(a) Is there any metalloporphyrins that could be useful for commercial purpose? Please highlight

Reply: Although great progress has been made, the comprehensive performances of porphyrin-based systems for water splitting and Zn-air battery are still weaker than those of noble metal systems. Therefore, the researches of porphyrin-based systems for electrochemical oxygen evolution reaction are still in the laboratory stage. Based on the available researches, cobalt porphyrin-based porous polymers which used as loading matrix or precursors for other catalysts could be potentially useful for commercial purpose. To be honest, the commercialization has been a long journey.

(b) What would be regeneration approaches for metalloporphyrins?

Reply: At present, most researches have conducted stability experiments, but few studies have focused on the regeneration issues for metalloporphyrins. It would be a very meaningful research topic in the future investigations. Considering metalloporphyrins could be degraded by light, heat, and bacterias, the carbonization of metalloporphyrins to prepare more stable catalysts (such as single-atom catalysts) maybe be an alternative approach to alleviate the problem.

(c) What are research gaps for commercial use?

Reply: There are still many problems to be solved for metalloporphyrins to be used as commercial catalysts for water splitting and Zn-air battery. The electrocatalytic performance and stability are at least two critical issues to be overcome. In the presence of various bacteria, different temperature and humidity ranges, long-term light radiation or electrolytic conditions, how to maintain high electrocatalytic performances and high stability at the same time is a difficult problem. This requires the unremitting efforts of more researchers. Together with (a) and (b), these opinions have been included in the revised version.

Round 2

Reviewer 2 Report

The authors have done a great job on the manuscript. The reviewer knows that a dedication statement is usually a single sentence at the beginning of a book or thesis. However, the reviewer does not know exactly where the authors could write the dedication statement on the paper. Is it would be in the acknowledgment section? please check. The reviewer is happy to recommend this paper for publication in this journal.

Best wishes!!